# Vortex Fermi Liquid and Strongly Correlated Quantum Bad Metal

Nayan Myerson-Jain,[1] Chao-Ming Jian,[2] and Cenke Xu[1]

[1]*Department of Physics, University of California, Santa Barbara, CA 93106*
[2]*Department of Physics, Cornell University, Ithaca, New York 14853, USA*
(Dated: September 22, 2022)

The semiclassical description of two-dimensional ($2d$) metals based on the quasiparticle picture suggests that there is a universal threshold of the resistivity: the resistivity of a $2d$ metal is bounded by the so called Mott-Ioffe-Regal (MIR) limit, which is at the order of $h/e^2$. If a system remains metallic while its resistivity is beyond the MIR limit, it is referred to as a "bad metal", which challenges our theoretical understanding as the very notion of quasiparticles is invalidated. The description of the system becomes even more challenging when there is also strong correlation between the electrons. Partly motivated by the recent experiment on transition metal dichalcogenides moiré heterostructure, we seek for understanding of strongly correlated bad metals whose resistivity far exceeds the MIR limit. For some strongly correlated bad metals, though a microscopic description based on electron quasiparticles fails, a tractable dual description based on the "vortex of charge" is still possible. We construct a concrete example of such strongly correlated bad metals where vortices are fermions with a Fermi surface, and we demonstrate that its resistivity can be exceptionally large at zero temperature. And when extra charge $\delta n_e$ is doped into the system away from half-filling, a small Drude weight proportional to $(\delta n_e)^2$ will emerge in the optical conductivity .

PACS numbers:

## I. INTRODUCTION

The most rudimentary description of a metal relies on the notion of quasiparticles, *i.e.* an electron near the Fermi surface can be well approximated as a wave packet between two consecutive elastic scatterings with impurities. This picture requires that $l_{\mathrm{m}} k_F > 1$, where $l_{\mathrm{m}}$ is the mean free path of the electrons from scattering with the impurities[1]. When $l_{\mathrm{m}} k_F \sim 1$, the resistivity of a two dimensional system is of the order of $h/e^2$, which is also known as the Mott-Ioffe-Regal (MIR) limit. The common wisdom is that, for noninteracting electrons, when the resistivity of a $2d$ metal exceeds the MIR limit, not only would the rudimentary description of the system fail, the system would actually become an insulator due to the Anderson localization. The potential metal-insulator transition (MIT) of a noninteracting $2d$ electron system[2] (within certain symmetry class such as the symplectic) should happen when the resistivity is of order $h/e^2$.

In strongly interacting electron systems, the universal threshold of resistivity $h/e^2$ still appears to hold. For electrons at half-filling (on average one electron per site) on a lattice, the competition between the interaction and kinetic energy can lead to an interaction-driven MIT between a metal and a Mott insulator phase. When the insulator is a particular type of spin liquid phase, this MIT can be understood through a parton construction[3,4], and the total resistivity follows the Ioffe-Larkin rule[5] $\rho = \rho_b + \rho_f$, where $\rho_b$ and $\rho_f$ are resistivity from the bosonic and fermionic partons respectively. $\rho_f$ is a smooth function across the MIT, and at low temperature $\rho_f$ mostly arises from disorder, which is expected to be small if we assume weak disorder. Hence at the MIT, the critical resistivity is mostly dominated by the bosonic parton $\rho_b$. The critical resistivity $\rho_b$ is expected to be of

order $h/e^2$ (though in the DC limit $\rho_b$ may acquire an extra factor of $7 \sim 8$, based on analytical evaluation in certain theoretical limit[6])[76].

Hence in most noninteracting as well as strongly interacting systems that we have understood, the resistivity of a $2d$ metallic system should be roughly bounded by the MIR limit. Hence if a $2d$ system remains metallic while its resistivity far exceeds the MIR limit, it challenges our theoretical understanding. These exotic metals are referred to as "bad metals"[1]. The recent experiment on transition metal dichalcogenides (TMD) revealed the existence of a novel interaction-driven MIT[7], where the universal MIR limit is violated: the DC critical resistivity at the MIT exceeds the MIR limit by nearly two orders of magnitude. The system is supposedly modelled by an extended Hubbard model of spin-1/2 electrons on a triangular moiré lattice[8,9], but the experimental finding is qualitatively beyond the previous theory of MIT. A few recent theoretical proposals[10,11] were made in order to understand this exotic MIT. The experiment mentioned above only revealed a critical point whose resistivity is clearly beyond the MIR limit. Given the current experimental finding and the strongly interacting nature of the system, it is natural to ask, *can there also be a stable bad metal phase of strongly correlated electrons, whose properties can be evaluated in a tractable way?*

In this work we discuss the construction of a *quantum bad metal state with longitudinal transport only*; the electrical resistivity $\rho_e$ of the state can far exceed the MIR limit even with weak disorder, at zero and low temperature. It is worth noting that the phenomenology of the state we construct is different from the original example of "bad metal" (hole doped cuprates) discussed in Ref. 1, where the resistivity increases with temperature monotonically and exceeds the MIR limit at high temper-

ature; while the resistivity of our "quantum bad metal" remains finite and large *at zero temperature*, and clearly violates the MIR bound. Our construction is formulated through the dual degrees of freedom of "charge vortex". The particle-vortex duality has a long history[12–14]. This duality was originally discussed for bosons, but recent developments have generalized the duality to fermion-vortex duality[15–19], as well as Chern-Simons matter theory to free Dirac or Majorana fermion duality[20–27]. And since the particle-vortex duality is still a "strong-weak" duality, when the charges are strongly correlated which invalidates a perturbative description based on quasiparticles, the vortices are weakly interacting through the dual gauge field, which facilitates a rudimentary description.

Hence one way to construct a quantum bad metal for charges is to drive the vortices into a good metal. The vortices can naturally form a good metal as long as (1) the vortex is a fermion, and (2) the fermionic vortices form a Fermi surface with a finite density of states. In the next section we will discuss how exactly a vortex becomes a fermion in our construction, and how to derive the charge responses of the system from the physics of vortices. We would like to clarify that we are not the first to investigate correlated electrons as vortex liquid. Besides the more well-known interpretation of composite fermions as "vortex liquid" in the context of half-filled Landau level (and similarly for charged bosons at filling 1)[15,28–31]; a metallic phase with *anomalously large* conductivity that emerges in amorphous thin film also motivated discussions of exotic physics of superconductor vortices[32–34]. We will compare our construction with the previous works.

## II. CONSTRUCTION OF THE QUANTUM BAD METAL

### A. General considerations

Before we detail our construction, some general considerations can already be made.

(1) As was pointed out in previous literatures, at least for charged bosons, the product of the conductivity of the charges and the conductivity of the vortices is a constant[35,36], *i.e.* $\sigma_e \sim 1/\sigma_v$. If this relation still (at least approximately) holds in our construction, it implies that if the vortex conductivity $\sigma_v$ follows the standard behavior of a good metal at finite temperature, then the resistivity $\rho_e(T)$ of charge should decrease with $T$, at least below certain characteristic energy scale.

(2) A charge vortex can generally be viewed as a point defect with circulating vorticity of charge current. A charge vortex must become an anti-vortex under spatial reflection $\mathcal{P}$. This is because the electric current circulation will reverse its orientation under reflection. If the vortices form a Fermi surface, in general it would break $\mathcal{P}$, as a Fermi surface usually is not invariant under the

particle-hole transformation. The same observation can be made for time-reversal $\mathcal{T}$: since charge density is invariant under $\mathcal{T}$, time-reversal would reverse the direction of electric current circulation. We will discuss later how to preserve $\mathcal{P}$ and $\mathcal{T}$ in our construction, *by enforcing certain particle-hole symmetry of the fermionic vortices.*

(3) As was pointed out in Ref. 28,29, the Wiedemann-Franz law $\kappa \sim T\sigma$ should generally be strongly violated in a vortex liquid, as the vortices carry entropy, but no charge. In the state we construct this is still true, the modified Wiedmann-Franz law should be $\kappa \sim LT\sigma_v \sim LT\rho_e$. The Lorenz number $L$ is about $\kappa/(T\sigma_e) \sim \rho_e^2$, which can be exceedingly larger than an ordinary metal. Here we remind the readers that our state has longitudinal transport only, and it can have both large resistivity and thermal conductivity.

(4) For a strongly interacting electron system, the relaxation of the electric current is pretty much independent from the relaxation of a single particle. Hence the physics of a strongly interacting electron liquid may be only captured by some hydrodynamical description without microscopic particles[37–42], as hydrodynamics is defined at a much larger length scale. But since the particle-vortex duality is a strong-weak duality, the interaction between electron density becomes

$$\sum_{i,j} V_{i,j} n_i n_j \sim \int d^2x \, \frac{1}{g} (\vec{\nabla} \times \vec{a})^2 \tag{1}$$

in the dual picture, where $g$ can be viewed as the gauge coupling of the gauge charges (vortices), and also the charge compressibility $\kappa_e$. The stronger the charge interaction is, the weaker is the bare gauge coupling of the vortices. The common "patch theory" for analyzing the RG flow of a Fermi surface coupled with a U(1) gauge field predicts that the gauge coupling would flow to a strongly coupled fixed point eventually[43–47]. But this patch theory breaks down when there is disorder, as disorder would mix different patches in the momentum space. But at least when the bare gauge coupling $g$ is weak enough (which corresponds to a strong charge density-density interaction), there should be a sufficient window for the gauge coupling to be viewed as a perturbation, and the momentum of the vortices can be transferred to the photons, and then relax through disorder before "feeding back" to the vortices. Hence in this sense *we can view the dual vortex system as an approximate vortex Fermi liquid.*

### B. Quantum Bad metal at half-filling on a lattice

The system we begin with is a strongly interacting electron system with half-filling (one electron per site on average) on a lattice, later we will discuss what happens when the system is doped away from half-filling. We

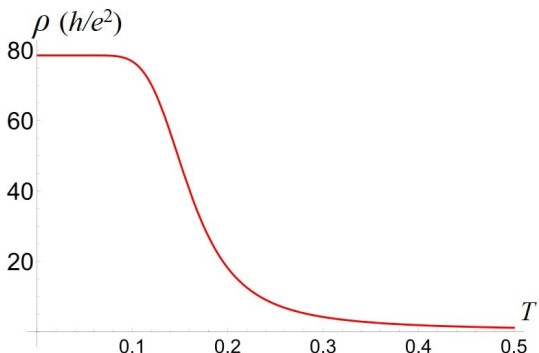

FIG. 1: The resistivity $\rho(T)$ as a function of temperature at low temperature, computed using the composition formula Eq. 11. We have chosen $\sigma_b = \exp(-\Delta_b/T)$ where $\Delta_b = 1$, and $\sigma_f \sim 200$. $\rho$ is measured in unit of $h/e^2$ in this plot. We can also give $\sigma_f$ a Fermi-liquid like temperature dependence, the plot remains qualitatively unchanged.

start with the standard "SU(2) slave rotor" theory for the electron operator[48–51]:

$$c_\uparrow = f_\uparrow z_1 - f_\downarrow^\dagger z_2^\dagger,$$

$$c_\downarrow = f_\downarrow z_1 + f_\uparrow^\dagger z_2^\dagger. \quad (2)$$

Here $(f_\uparrow, f_\downarrow)$ is a fermionic spinon doublet (fermionic partons) that carries spin-1/2, and $(z_1, z_2)$ are slave rotors (bosonic partons) carrying the electric charge. This formalism can maximally host a SU(2) charge transformation and a SU(2) gauge transformation, and both transformations can be made explicit by rewriting Eq. 2 in a matrix form (see *e.g.* Ref. 52,53 and references therein)[77]. But for our purpose it suffices to assume all SU(2) transformations, including the spin symmetry are broken down to U(1). In fact, on a lattice with frustration, both the charge SU(2) and the gauge SU(2) transformations are broken down to U(1) by the most natural mean field states of $z_\alpha$ and $f_\alpha$. The assignment of the electric charge symmetry U(1)$_e$, gauge symmetry U(1)$_g$ and spin symmetry U(1)$_s$ on the partons is

$$\text{U}(1)_e : z_1 \to e^{i\theta_e} z_1, \quad z_2 \to e^{-i\theta_e} z_2, \quad f_\alpha \to f_\alpha;$$

$$\text{U}(1)_g : z_1 \to e^{i\theta_g} z_1, \quad z_2 \to e^{i\theta_g} z_2, \quad f_\alpha \to e^{-i\theta_g} f_\alpha,$$

$$\text{U}(1)_s : z_\alpha \to z_\alpha, \quad f_\uparrow \to e^{i\theta_s} f_\uparrow, \quad f_\downarrow \to e^{-i\theta_s} f_\downarrow. \quad (3)$$

The system being at half-filling implies that the total rotor number of $z_1$ and $z_2$ are equal: $\sum_i n_{1,i} = \sum_i n_{2,i}$. There is a dynamical U(1)$_g$ gauge field $a_\mu$ that couples to both $z_\alpha$ and $f_\alpha$. The U(1) gauge constraint demands that on every site $i$, $\sum_\alpha n_{\alpha,i} + 1 = \sum_\alpha f_{\alpha,i}^\dagger f_{\alpha,i}$. A detailed discussion of the physical meaning of the partons introduced can be found in Ref. 52.

This slave rotor parton construction allows us to construct many states of the strongly interacting electron system which are difficult to visualize using free or weakly interacting electrons. For example, if the bosonic partons $z_\alpha$ are in a trivial bosonic Mott insulator state with $n_{1,i} = n_{2,i} = 0$ (meaning $\sum_\alpha f_{\alpha,i}^\dagger f_{\alpha,i} = 1$ on each site), the system becomes a Mott insulator of electrons with a charge gap, and the spin physics of the Mott insulator depends on the state of $f_\alpha$. Various spin liquid states can be designed and classified depending on the mean field band structure of $f_\alpha$[54].

The many-body state of electrons is determined by the states of the partons. In the last decade the study of symmetry protected topological states (SPT) significantly broadened our understanding of the states of matter[55,56], which also allows us to construct even more novel states of electrons using the partons. We first use $z_\alpha$ to define two other composite bosonic fields: $\phi_e = z_1^\dagger z_2$, $\phi_g = z_1 z_2$. $\phi_e$ and $\phi_g$ carry charge $(2,0)$ and $(0,2)$ under $(\text{U}(1)_e, \text{U}(1)_g)$. Then we drive the composite fields $\phi_e$ and $\phi_g$ into a bosonic SPT state with U(1)$_e$ and U(1)$_g$ symmetries, which is the bSPT state for two flavors of bosons constructed in Ref. 57. The physics of this bSPT state is analogous to the quantum spin Hall insulator: the vortex of $\phi_e$ carries charge of $\phi_g$, and vice versa. If we follow the Chern-Simons description of the bSPT[58], this state is

$$\mathcal{L}_{\text{bSPT}} = \frac{iK^{IJ}}{2\pi} \tilde{a}_I \wedge d\tilde{a}_J + \frac{i2}{2\pi} \tilde{a}_1 \wedge da + \frac{i2}{2\pi} \tilde{a}_2 \wedge dA^e, \quad (4)$$

where $K^{IJ}$ takes the same form as the Pauli matrix $\sigma^x$. Here $*d\tilde{a}_1$ and $*d\tilde{a}_2$ are the dual of the currents of $\phi_g$ and $\phi_e$ respectively. This bSPT state of the rotors also has a particle-hole symmetry of the bosonic rotors, and in this bSPT state the expectation value of the total rotor number of both $z_1$ and $z_2$ is zero. Please note that the total rotor number being zero does not imply a trivial vacuum state, as the rotor number (just like a spin $S^z$ operator) can take both positive and negative values.

A bSPT state is gapped, and also nondegenerate, hence it is safe to integrate out the bosonic degree of freedom, and obtain the response to the gauge fields. After integrating out $\tilde{a}_{1,2}$ from Eq. 4, we obtain:

$$\mathcal{L} = \mathcal{L}_F(f_\alpha, a_\mu) + \frac{4i}{2\pi} a \wedge dA^e + \frac{1}{g}(\vec{\nabla} \times \vec{a})^2 + \cdots. \quad (5)$$

One can also introduce the external gauge field for the spin symmetry U(1)$_s$, but it won't have a nontrivial response from the bSPT state.

The mutual Chern-Simons term in the last term of Eq. 5 fundamentally changes the physics of the system in the following way:

The electric charge current, which is defined as $J^e = \delta\mathcal{L}/\delta A_e$, is identified as

$$J^e = e\frac{4}{2\pi} * da, \quad (6)$$

meaning the flux of $a_\mu$ now carries electric charge $4e$. Hence the bSPT state so constructed turns the gauge

field $a_\mu$ into the dual of the charge current in the sense of the particle-vortex duality[12–14], and turns the gauge charge of $a_\mu$ into the charge vortex. *The fermionic parton $f_\alpha$, which carries gauge charge 1 under $a_\mu$, now automatically becomes the vortex of the electric charge*, as when a charge (now the flux of $a_\mu$) circulates the gauge charged $f_\alpha$, it would accumulate a Berry's phase. It is worth noting that, for more general bSPT states of $z_\alpha$ with only mutual $U(1)_e \times U(1)_g$ Chern-Simons response, the electric charge carried by the flux of $a_\mu$ has to be an integer multiple of $4e$. Hence, the bSPT described above is the minimal non-trivial choice.

Now we take the long wavelength limit, and integrate out both $f_\alpha$ and $a_\mu$; we also choose the temporal gauge with $a_\tau = 0$. The response Lagrangian in terms of $A^e$ is

$$\mathcal{L}_{\text{res}} = \sum_\omega \frac{1}{2} \frac{4}{\pi^2} \frac{\omega^2}{\Pi_f(\omega)} |\vec{A}^{e,t}(\omega)|^2. \qquad (7)$$

$\vec{A}^{e,t}$ is the transverse component of $\vec{A}^e$. $\Pi_f(\omega)$ is the polarization of $f_\alpha$, and it should be proportional to $i\omega\sigma_f(\omega)$ after analytic continuation[59], where $\sigma_f(\omega)$ is the conductivity of the fermionic parton (*i.e.* also the vortex) $f_\alpha$. This implies that the electrical resistivity of the system should be

$$\rho_e(\omega) = \frac{\pi^2}{4}\sigma_f(\omega) = \frac{\pi^2}{4}\left(\frac{\sigma_0}{1 - i\omega\tau_v}\right). \qquad (8)$$

We note that here $\rho_e$ is measured with unit $\hbar/e^2$; $\sigma_f$ is computed in the convention that $f_\alpha$ carries charge 1. Here we can evaluate the conductivity of the good metal of $f_\alpha$ using the rudimentary Drude formula. $\sigma_0$ is the conductivity of the $f_\alpha$ in the DC limit. It was shown recently that even when a Fermi surface is coupled to a dynamical gauge field, the response to the gauge field is still exactly the same as what is computed by the Drude theory (at least when there is no disorder)[60]. We also exploit the fact that, when the electron density has a strong interaction, the bare gauge coupling becomes weak, and the photon-vortex interaction remains perturbative at least within a large window of scale. An analysis of the fermions interacting with gauge field in a disordered environment can be found in Ref. 61.

In Eq.8, $\sigma_0$ can be rather large, namely the vortices form a good metal, when there is a finite Fermi surface of $f_\alpha$ and the disorder is weak. In this case the electrical resistivity of the system can be far beyond the MIR limit, *i.e.* the system is a very bad metal.

Now we investigate the spatial reflection symmetry $\mathcal{P}$ of our system. And let us use $\mathcal{P}_y : y \to -y$ as an example. We assume that $c_\alpha$ changes up to a sign under $\mathcal{P}_y$, then this leads to the transformation of $f_\alpha$, $z_\alpha$:

$$\mathcal{P}_y: \quad f_\alpha \to (\sigma^x)_{\alpha\beta}f_\alpha^\dagger, \quad z_\alpha \to (\sigma^x)_{\alpha\beta}z_\beta^\dagger. \qquad (9)$$

The $U(1)_e$ and $U(1)_g$ charges are even and odd under $\mathcal{P}_y$ respectively. This means that $\tilde{a}_{1,2}$ transform oppositely under $\mathcal{P}$, and the bSPT state preserves $\mathcal{P}$ based on Eq. 4. In order to ensure the reflection symmetry, we also need the band structure of $f_\alpha$ to satisfy $\varepsilon_\uparrow(k_x, k_y) = -\varepsilon_\downarrow(k_x, -k_y)$.

Our construction also preserves a (special) time-reversal symmetry $\mathcal{T}$ defined as following: the electron operator is still invariant under $\mathcal{T}$ (up to an extra sign). We can choose the following transformations of $z_\alpha$ and $f_\alpha$ to ensure the desired transformation of the electrons:

$$\mathcal{T}: \quad z_\alpha \to (\sigma^x)_{\alpha\beta}z_\beta^\dagger, \quad f_\alpha \to (\sigma^x)_{\alpha\beta}f_\beta^\dagger. \qquad (10)$$

As we can see, the $U(1)_e$ and $U(1)_g$ charges are again even and odd under $\mathcal{T}$ respectively. This means that $\tilde{a}_{1,2}$ transform oppositely under $\mathcal{T}$, and the bSPT state preserves $\mathcal{T}$ based on Eq. 4. In order to ensure the time-reversal symmetry, we also need the band structure of $f_\alpha$ to satisfy $\varepsilon_\uparrow(\vec{k}) = -\varepsilon_\downarrow(-\vec{k})$. More precisely, here the time-reversal is a product between the particle-hole transformation and a spatial-inversion[78]. A more realistic time-reversal symmetry for electrons with $\mathcal{T}^2 = -1$ can be defined and preserved if we introduce another orbital flavor to the electrons.

So far we have ignored the conductivity of the bosonic partons, which is valid when the energy scale is much smaller than the gap of the bosons $\Delta_b$. With finite frequency $\omega$, the bosons will also make two new nonzero contributions to the longitudinal response of $A^e$. The first of which is a simple addition to the response Lagrangian of the boson polarization *i.e.* $\mathcal{L}_{\text{res}}^b \sim \frac{1}{2}\Pi_b(\omega)|\vec{A}^{e,t}(\omega)|^2$, for which $\Pi_b(\omega) \to i\omega\sigma_b(\omega)$ after analytic continuation. This only modifies the charge conductivity by shifting it a value of $\sigma_b(\omega)$. The second, more interesting contribution, is that a longitudinal term will be generated for the internal gauge field $a_\mu$ as well. In principle $\Pi_b(\omega)$ can be different for $A_\mu^e$ and $a_\mu$, but without loss of generality we assume that they are the same. Since the internal gauge field and electromagnetic gauge field transform differently under $\mathcal{P}$ and $\mathcal{T}$ and so do the $\phi_g$ and $\phi_e$ current, there cannot be mixed terms between $a$ and $A^e$ that would lead to mutual longitudinal response in the effective Lagrangian. Then eventually the electrical conductivity of the system follows the following composition rule

$$\sigma_e = \sigma_b + \frac{4}{\pi^2}\left(\frac{1}{\sigma_f + \sigma_b}\right). \qquad (11)$$

In this equation, $\sigma_e$ is measured in the unit of $e^2/\hbar$; $\sigma_b$ and $\sigma_f$ are computed in the convention of charge 1 and $\hbar = 1$. This composition is very different from the Ioffe-Larkin rule[5], and we expect this composition rule to be valid for temperature far smaller than the gap of the bosonic parton, *i.e.* when the thermally activated bosonic partons are very dilute. In Fig. 1 we plot $\rho_e(T) = 1/\sigma_e(T)$ measured in unit of $h/e^2$ using the composition rule Eq. 11.

Another quantity of interest is the charge compressibility. The compressibility can be computed from the charge density-density correlation function which can be

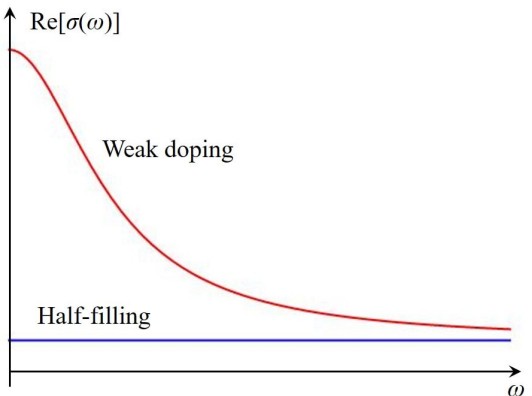

FIG. 2: The optical conductivity $\text{Re}[\sigma(\omega)]$ near $\omega = 0$ at half-filling, and with small doping $\delta n_e$.

attained by reading off the $A_\tau(q)A_\tau(-q)$ term in $\mathcal{L}_{\text{res}}$ at vanishing frequency after integrating out all the matter and the internal gauge fields. As we discussed before, one contribution to the compressibility is proportional to the gauge coupling $g$, which is evident after integrating out $\vec{a}$ from Eq. 5. Eventually the compressibility should involve the bare charge density-density interaction, after being renormalized by integrating out the fermions $f_\alpha$. In the limit $\omega \to 0$ and large gap of bosonic rotors, the total compressibility is given by

$$\kappa_e(\mathbf{q}) = \frac{1}{(\kappa_0)^{-1} + \frac{\pi^2}{4}\frac{\Pi_f(q)}{|\mathbf{q}|^2}}, \quad (12)$$

where $\kappa_0$ is the "bare" compressibility of the system when the gauge field $a_\mu$ is not coupled to any matter fields, hence $\kappa_0$ is proportional to the gauge coupling $g$: $\kappa_0 = g(4/\pi^2)$. If we choose a simple quadratic dispersion or a circular Fermi surface for the vortices, then the result for the fermionic vortex polarization at zero frequency is well-known to be[59,62] $\Pi_f(\mathbf{q}) \sim |\mathbf{q}|^2/(12\pi m_v)$ where $m_v$ is the effective mass of the fermionic vortices. Like the charge conductivity, this gives us a composition rule for the charge compressibility in terms of linear response functions for the different species of partons. For the more general case of a finite boson gap, this composition rule involves both the boson compressibility and boson polarization $\Pi_b \sim \chi_b|\mathbf{q}|^2$:

$$\kappa_e(\mathbf{q}) = \kappa_b(\mathbf{q}) + \frac{1}{(\kappa_0)^{-1} + (\chi_b + \chi_f)\pi^2/4}. \quad (13)$$

Here, $\chi_b$ is the magnetic susceptibility of the bosons. We expect this composition rule to hold at small finite temperature much below the boson gap.

## C. Physics at weak doping

We would also like to consider the effects of weak charge doping $\delta n_e$ away from one electron per site. As the charge density is now bound with the internal gauge flux through the mutual CS term in Eq. 5, weakly doping the system corresponds to adding a background $U(1)_g$ gauge field with small average magnetic flux. Since the vortices (spinon $f_\alpha$) form a good metal, their conductivity may still be evaluated through the semiclassical Drude theory. Using the semiclassical equation of motion with magnetic field, it is straight forward to compute the Drude conductivity of $f_\alpha$:

$$\sigma_f(\omega) = \left(\frac{\frac{1}{\tau_v} - i\omega}{\omega_c^2 + (\frac{1}{\tau_v} - i\omega)^2}\right)\frac{1}{\tau_v}\sigma_0. \quad (14)$$

$\omega_c$ is the cyclotron frequency of the vortices $\omega_c = \bar{b}/m_v$, and $\bar{b} \sim \delta n_e$ is the average flux seen by the fermionic vortices. Note that the conductivity $\sigma_f(\omega)$ of $f_\alpha$ only contains a longitudinal component given by the expression above. The Hall conductivities from $f_\uparrow$ and $f_\downarrow$ cancel each other due to the time-reversal symmetry $\mathcal{T}$ which involves a particle-hole transformation of the spinons $f_\alpha$. From the relation between the charge conductivity and vortex conductivity in Eq. 11, if the boson gap is taken to infinity, we can extract the main interesting piece of the charge conductivity, which is given by

$$\sigma_e(\omega) \sim \frac{4}{\pi^2\sigma_f(\omega)} = \frac{4\tau_v}{\pi^2\sigma_0}\left[\frac{\omega_c^2}{\frac{1}{\tau_v} - i\omega} + \left(\frac{1}{\tau_v} - i\omega\right)\right]. \quad (15)$$

At zero doping, the optical conductivity does not have a Drude weight; *but the presence of added charge has introduced a new Drude peak to the optical conductivity with Drude weight:*

$$D \sim \frac{2\omega_c^2\tau_v}{\pi^2\sigma_0} \sim (\delta n_e)^2, \quad (16)$$

and the Drude weight is proportional the square of the doped charge density, contrary to the ordinary Drude theory where the Drude weight is linear with the charge density.

The DC resistivity now takes the form of

$$\rho_e = 1/\sigma_e(0) = \frac{\pi^2\sigma_0}{4}\frac{1}{1 + \tau_v^2\omega_c^2}. \quad (17)$$

The Lorenz number, defined as $L = \kappa/(T\sigma_e)$ becomes

$$L = \frac{\kappa}{T\sigma_e} \sim \frac{L_0\sigma_f}{\sigma_e} \sim \frac{\pi^2}{4}\frac{L_0\sigma_0^2}{(1 + \tau_v^2\omega_c^2)^2}. \quad (18)$$

Here $\kappa$ represents the thermal conductivity. Both the resistivity, and Lorenz number decrease with the doped charge density. Combing with the emergence of the Drude weight under doping, it suggests that doping would eventually drive the the system more like a normal metal. Here we have ignored the thermal conductivity arising from the gauge bosons, which also transports heat without charge, hence it also contributes to the violation of the Wiedemann-Franz law.

### D. Nearby Phases

#### (1) *Spin liquid Mott insulator, and metal*

The quantum bad metal state constructed above can be driven to a Mott insulator which is also a U(1) spin liquid with a Fermi surface of spinon $f_\alpha$, by driven $(z_1, z_2)$ into a trivial bosonic Mott insulator, with zero rotor number of $z_1$ and $z_2$. In the Mott insulator, $f_\alpha$ still has Fermi pockets, but the gauge flux of $a_\mu$ no longer carries any nontrivial quantum number. This is one of the most studied spin liquid states in the literature[3,63,64], with potential applications to a variety of materials.

The quantum phase transition between a bSPT state and a trivial Mott insulator of the boson is described by the $N = 2$ QED[58,65], and this theory is part of a web of duality involving also the easy-plane deconfined quantum critical point[18,20,66–69]. The original theory of the bSPT-MI transition is now coupled to an extra dynamical gauge field $a_\mu$. When there is no disorder, the dynamics of $a_\mu$ is overdamped by the Fermi surface of $f_\alpha$, then we do not expect the gauge field $a_\mu$ to change the infrared fate of the transition. The presence of disorder may complicate the nature of the bSPT-MI transition.

The quantum bad metal phase can also be driven into an ordinary metal phase by condensing either $z_1$ or $z_2$. The U(1)$_g$ gauge field will be gapped out by the Higgs mechanism, and the spinon operator $f_\alpha$ becomes the electron operator due to the condensate of, *e.g.* $z_1$, similar to the previous theory of interaction-driven MIT[3,4].

#### (2) *Charge-4e superconductor*

Starting with the quantum bad metal, we can also drive the spinon $f_\alpha$ into a trivial insulator without special topological response, likely through a Lifshitz transition where the Fermi pockets shrink to zero. Then the action of $a_\mu$ is just the ordinary Maxwell term, which describes a photon phase. The monopole which creates and annihilates the gauge flux is prohibited here as the gauge flux carries charge-4e as we discussed before, and the electric charge is a conserved quantity. The photon phase of the gauge field $a_\mu$ is also dual to the condensate of its flux, *i.e.* a condensate of charge$-4e$, or in other words a charge$-4e$ superconductor.

#### (3) *$Z_2$ spin-charge topological order*

We can also consider a situation where the fermionic vortices $f_\alpha$ form a "superconductor", *i.e.* the Cooper pair of $f_\alpha$ condenses. This condensation will gap out $a_\mu$ through a Higgs mechanism, and break $a_\mu$ down to a $Z_2$ gauge field, which supposedly forms a $Z_2$ topological order. Like all the $Z_2$ topological orders, here there are three types of anyons with mutual semionic statistics. One type of anyon is $f_\alpha$, another is the half-flux of $a_\mu$. Since the flux of $a_\mu$ carries charge-4e, the half-flux of $a_\mu$ carries charge-2e.

### E. Other Constructions

One can also construct a similar quantum bad metal phase starting with a charge-2e spin-singlet superconductor. Let us assume there are two flavors of bosons, $b_1$ and $b_2$, which carry charge $\pm 2e$ respectively. We take the following parton treatment for $b_\alpha$:

$$b_1 = \psi_1 f, \quad b_2 = \psi_2 f, \qquad (19)$$

where all the partons are complex fermions. Apparently there is also a gauge U(1)$_g$ shared by the partons. $\psi_{1,2}$ carries electric charge $\pm 2$, and gauge charge $+1$ of a dynamical internal gauge field; $f$ carries gauge charge $-1$ of the internal gauge field. We now consider the following state: $f$ again forms a band structure which is a good metal; $\psi_\alpha$ forms a quantum "psudospin" Hall insulator, in the sense that the flavor index of $\psi_\alpha$ is viewed as a pseudospin index. After integrating out $\psi_\alpha$, a mutual Chern-Simons term between the external EM field $A^e$ and the internal gauge field $a$ is generated, with the same form as Eq. 5. We assume that there is no other conserved charges other than the charge U(1). The charge response of this construction can be evaluated following the steps of the previous section.

The bSPT state is one of the states that the bosonic partons $(z_1, z_2)$ can form that make the charge vortex a fermion. There are other options which achieve the similar effect, if we allow topological degeneracy. For example, $z_1$ and $z_2$ can each form a bosonic fractional quantum Hall state with Hall conductivity $\pm 1/(2k)$ where $k$ is an integer. Although there is a bit subtlety of integrating out a topological order, suppose we can do this, the response mutual CS theory in Eq. 5 would have level $2/k$. The rest of the discussion follows directly.

We would like to compare our state with other "vortex liquids" discussed in previous literature, for example the well-known Dirac vortex liquid in the context of half-filled Landau level[15,29]. The electrical conductivity tensor of the system reads $\sigma_{ij} = \delta_{ij}\sigma^* + \frac{e^2}{2h}\epsilon_{ij}$, where $\sigma^* \sim 1/\sigma_v$, and $\sigma_v$ is the conductivity of the Dirac composite fermions (the vortices). Although the longitudinal conductivity of the system can be small when $\sigma_v$ is large, the longitudinal resistivity would still be small due to the nonzero Hall conductivity. Hence in the simplest experimental set-up where the transport is measured along the $\hat{x}$ direction while the $\hat{y}$ direction of the sample has an open boundary, the measured longitudinal resistivity along the $x$ direction would be small.

We can also put $c_\uparrow$ in the Dirac vortex liquid of the half-filled Landau level; and $c_\downarrow$ in the time-reversal conjugate state of $c_\uparrow$. In this case in order to correctly extract the longitudinal electrical resistivity, we need to introduce both the external electromagnetic field, and a "spin gauge field" $A^s$:

$$\mathcal{L} = \mathcal{L}_D(\psi_1, a_1) + \frac{\mathrm{i}}{4\pi} a_1 \wedge A_1 + \frac{\mathrm{i}}{8\pi} A_1 \wedge dA_1,$$

$$+ \mathcal{L}_D(\psi_2, a_2) - \frac{\mathrm{i}}{4\pi} a_2 \wedge A_2 - \frac{\mathrm{i}}{8\pi} A_2 \wedge dA_2. \quad (20)$$

$\psi_{1,2}$ are the composite Dirac fermions, and their Lagrangian $\mathcal{L}_D$ should in general have a nonzero chemical potential. We have introduced two external gauge fields $A_1 = A^e + A^s$, and $A_2 = A^e - A^s$. The system does not have a net charge Hall response, but there is a spin-Hall effect, $i.e.$ there is a mutual CS term between the electromagnetic field and the spin gauge field. The existence of the spin Hall response will again lead to a small longitudinal electrical resistivity when $\psi_{1,2}$ are good metals. By contrast, in our construction presented in the previous section, there is only longitudinal transport, hence a large vortex conductivity would ensure a large longitudinal electrical resistivity.

Another vortex liquid with fermionic vortex was discussed in Ref. 32, aiming to understand the observed metallic state with an anomalously large conductivity between the superconductor and insulator in amorphous thin films. There the vortex is turned into a fermion through manual flux attachment.

Vortices can also play an important role in quantum magnets. Exotic quantum spin liquid states were also constructed through fermionic spin vortices in previous literature[70–74]. These works generally use two approaches to generate fermionic vortices: one can either introduce fermionic partons for the vortex operator, or turn a vortex into a fermion through flux attachment when the vortex sees a background magnetic field (dual of fractional spin density). In our work, instead of granting existing vortices fermionic statistics, an interpretation of the fermionic partons as charge vortices naturally emerges due to the topological physics of the bosonic sector.

## III. SUMMARY

We present a construction of a strongly interacting quantum bad metal phase, $i.e.$ at zero and low temper-ature the resistivity is finite but exceedingly larger than the MIT limit $h/e^2$, by making the charge vortices a good metallic phase with a vortex Fermi surface. In this construction the charge vortex is naturally a fermion by driving the charge degree of freedom into a bosonic symmetry protected topological state. The quantum bad metal so constructed has the following features: (1) its resistivity can be exceedingly larger than the MIR limit; (2) a small Drude weight proportional to $(\delta n_e)^2$ emerges under weak charge doping away from half-filling (one electron per unit cell); (3) like previously discussed vortex liquids, our construction should also have strong violation of the Wiedemann-Franz law. We also demonstrated that this quantum bad metal phase is next to a charge-$4e$ superconductor, a $Z_2$ spin-charge topological order, the Mott insulator phase which is also a well-studied spin liquid, and a normal metal phase.

The exoticness of the state we constructed is of "quantum nature", as strictly speaking a bSPT state that our construction strongly relies on is only sharply defined at zero temperature. At high temperature all the partons will be confined, and our state no longer enjoys a tractable description in terms of the dual weakly interacting vortices. Hence our state is different from the original example of bad metal discussed in Ref. 1, $i.e.$ the cuprates materials with hole doping, where the resistivity of the system in each $2d$ layer reaches the threshold of bad metal at finite temperature.

The authors thank Matthew Fisher and Steve Kivelson for very helpful discussions. We also thank Umang Mehta and Xiao-Chuan Wu for participating in the early stage of the work. This work is supported by the NSF Grant No. DMR-1920434, and the Simons Investigator program. C.-M. J. is supported by a faculty startup grant at Cornell University

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

[33] J. Wu and P. Phillips, Phys. Rev. B **73**, 214507 (2006), URL https://link.aps.org/doi/10.1103/PhysRevB.73.214507.

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
