# Peer review of "Vortex Fermi Liquid and Strongly Correlated Quantum Bad Metal"

_SciPost Physics_

## Round 2 · Referee Report · Anonymous (Referee 2) · 2023-1-9

Report

This manuscript deals with a well studied problem in the literature, which the authors acknowledge by citing most important references in the field. Concerning the vortex liquid, the authors also acknowledge that "we are not the first to investigate correlated electrons as vortex liquid". The duality approach employed is also standard. At first sight, the manuscript does not appear as something involving great originality. There are nevertheless a few SciPost criteria of acceptance that may be considered, like for example, "open a new pathway in an existing or a new research direction, with clear potential for multipronged follow-up work". Before making a final decision, I would like the authors address a few points, which are most likely easy to comment/answer.

  1. It seems there is a typo in Eq. (4), the mutual Chern-Simons term (with the K-matrix) has to be divided by $4\pi$ rather than $2\pi$, otherwise the charge will not be $4e$ as stated. As it stands, integrating out $\widetilde{a}_1$ leads to the constraint $\widetilde{a}_2=-a$ and we are left with $-\frac{2}{2\pi} a\wedge dA^e$.

  2. Since the authors don't write the fermionic Lagrangian from Eq. (5) explicitly, it would be useful for the reader if they could provide more information about it. After all the authors integrate out the fermions later on to obtain the effective Lagrangian (which is actually an action) of Eq. (7). Incidentally, why are the authors not writing the momentum dependence of the vacuum polarization at this stage? I understand that the calculation of the conductivity requires letting the momentum vanish, but is there any particular reason why the momentum is not being written? Is it because of the nature of the background field?

  3. The authors mention in the conclusion that the partons are confined at higher temperatures, so parton deconfinement is happening as a quantum phase transition at zero temperature. However, there are examples of deconfinement by temperature effects. For instance, quarks deconfine at very high temperatures. An analytically tractable example of deconfinement at finite temperture is Polyakov's compact electrodynamics in 2+1 dimension, where test charges are permanently confined at zero temperature, but deconfine at finite temperature, see for instance the old paper by B Svetitsky, LG Yaffe, at https://doi.org/10.1016/0550-3213(82)90172-9 . What is the role of the charge $4e$ in this case? Is this the reason preventing finite temperature deconfinement to happen? In several condensed matter theories the finite temperature deconfinement does not happen because the theory becomes more classical. The authors seem to imply that this is the case, which seems plausible.

---

## Round 2 · Referee Report · Anonymous (Referee 1) · 2023-1-9

Strengths

1- tractable model with clear calculations. 2- addressing the important problem of understanding how bad metal type behavior can arise.

Weaknesses

The below is not a true "weakness" yet an item that the authors might consider. It is a matter of style yet the authors may wish to add some further self-contained physical background and explanations before briefly citing results of earlier references and jotting down results from these (such as in, e.g., Eq. 4).

Report

This paper is very interesting and I strongly recommend its publication.

Finding a physically appealing tractable model violating the Mott-Ioffe-Regel bound is a long standing quest. The main impetus for this problem has been the "bad metal" behavior associated with a violation of this bound at high temperatures. The work by Nayan Myerson-Jain, Chao-Ming Jian, and Cenke Xu makes a notable related advance in illustrating how violations may arise at zero temperature. A main workhorse is a simple analysis of fermionic dual vortices in a parton construction. The reasoning used by the authors is elegant.

---

## Editorial Decision

awaiting_resubmission